# First Report of Triple Viral Co-Infection (PPV, PCV2, PCMV) in Wild Boars in the Western Balkans

**DOI:** 10.3390/pathogens14070710

**Published:** 2025-07-18

**Authors:** Dimitrije Glišić, Sofija Šolaja, Kukilo Stevan, Vesna Milićević, Miloš Vučićević, Jelena Aleksić, Dajana Davitkov

**Affiliations:** 1Department of Virology, Institute of Veterinary Medicine of Serbia, Janisa Janulisa 14, 11000 Belgrade, Serbia; sofija.solaja@nivs.rs (S.Š.); vesna.milicevic@nivs.rs (V.M.); 2Veterinary Ambulance “One Health”, 78250 Laktasi, Bosnia and Herzegovina; kukilovet90@gmail.com; 3Faculty of Veterinary Medicine, University of Belgrade, Bulevar Oslobođenja 18, 11000 Belgrade, Serbia; vucicevic@vet.bg.ac.rs (M.V.); alexjellena@vet.bg.ac.rs (J.A.); dajana@vet.bg.ac.rs (D.D.)

**Keywords:** wild boar, porcine parvovirus (PPV), porcine circovirus type 2 (PCV2), porcine cytomegalovirus (PCMV), African swine fever virus (ASFV), classical swine fever virus (CSFV), pseudorabies virus (PRV), Serbia, Republic of Srpska

## Abstract

Wild boars are recognized reservoirs of numerous viral pathogens, posing a significant risk to domestic pig populations, particularly in areas with poor biosecurity. This study assessed the prevalence and co-infection patterns of porcine circovirus type 2 (PCV2), porcine parvovirus (PPV), porcine cytomegalovirus (PCMV), African swine fever virus (ASFV), classical swine fever virus (CSFV), and pseudorabies virus (PRV) in wild boars from western Serbia and the Republic of Srpska (Bosnia and Herzegovina). Sixty-six spleen samples from legally hunted wild boars were analyzed by qPCR. All animals were negative for ASFV, CSFV, and PRV. The cumulative prevalence of infection with at least one of the other three viruses was 86.4% (95% CI: 76.2–92.8%). PCMV was detected in 74.2% of samples, PCV2 in 50%, and PPV in 28.8%. Co-infections were common: 42.4% of animals were positive for two viruses, and 12.1% for all three. A statistically significant association was observed between triple co-infection and sex, with higher rates in males. Subadult wild boars showed the highest PCV2 + PCMV co-infection rate (*p* = 0.0547). These findings highlight the need to expand molecular surveillance, particularly for PCMV, in both wild and domestic pigs, especially in regions reliant on low-biosecurity backyard farming.

## 1. Introduction

Wild boars are an autochthonous species on the Balkan Peninsula and serve as reservoirs for numerous viral diseases [1]. By maintaining these pathogens within their populations, they complicate disease eradication efforts. Wild boars have been implicated in transmitting significant viral diseases, such as African and Classical swine fever (ASF and CSF), which can severely impact the pig industry [2]. Additionally, they carry other viral pathogens that, although less fatal, still cause substantial economic losses for both backyard and industrial farming operations [3]. In Serbia and Bosnia and Herzegovina, most farms are small, family-owned backyard operations with minimal or no biosecurity measures. Because of this, they frequently serve as entry points for infectious diseases, acting as stepping stones for pathogens to spread into larger industrial pig farms [4]. Porcine parvovirus (PPV) causes reproductive disorders in pigs [5]. It has a single-stranded DNA (ssDNA) and belongs to the family *Parvoviridae*, which comprises eight distinct porcine parvovirus species (PPV1–PPV8) distributed across several genera [6]. Porcine circoviruses are small viruses containing ss-circular DNA. They belong to the genus *Circovirus* and the family *Circoviridae* [7]. So far, four porcine circovirus types (PCV1–4) have been described. Porcine circovirus type 2 (PCV2) is commonly found in pigs and can cause several conditions, including postweaning multisystemic wasting syndrome, porcine dermatitis, and nephropathy syndrome, as well as reproductive and respiratory disorders [7]. Pseudorabies virus (PRV) causes fatal neurological disease in piglets, respiratory disorders in fattening pigs, and reproductive problems in sows [8]. It can also infect other animal species, leading to severe and often lethal neurological symptoms. According to the International Committee on Taxonomy of Viruses, PRV belongs to the genus *Varicellovirus* within the family *Herpesviridae* [9]. Porcine cytomegalovirus or porcine roseolovirus (PCMV/PRV) is a double-stranded DNA virus classified within the genus *Roseolovirus*, family *Herpesviridae* [9]. PCMV is considered a ubiquitous pathogen in domestic pigs and wild boars, typically causing mild or no clinical signs. However, in naïve sows, it may occasionally lead to reproductive and respiratory issues or even mortality [10]. ASF is a fatal viral hemorrhagic disease affecting members of the family *Suidae*. It is caused by a linear, double-stranded DNA virus classified in the genus *Asfivirus*, within the family *Asfarviridae* [11]. Swine pestiviruses include several of the most significant viral pathogens affecting the pig industry, such as CSF (*Pestivirus suis*), atypical porcine pestivirus (APPV/*Pestivirus scrofae*), Bungowannah virus (*Pestivirus australiaense*), and Linda virus (*Pestivirus* L.) [12]. Clinical signs vary and may include neurological symptoms in piglets, as well as a hemorrhagic syndrome in cases of CSF that closely resembles ASF [13].

Viral diseases in wild boars have been the focus of several studies in Serbia. High prevalence rates of PCV2 have been reported, reaching up to 69.9%, while PRV has been detected in 18–20% of tested animals [3,14,15]. PCMV was identified in a study by Jezdimirović et al., with a reported prevalence of 4.8% [14]. Pestiviruses have also been found in wild boar populations, with APPV last detected in 2015 [16]. As for CSF, the most recent case in Serbia occurred in 2010 [17]. Following that, a strict vaccination campaign was launched, and no new cases have been reported in the past decade. ASF has been present in the wild boar population in Serbia since 2020, with new cases reported each year. Both the number of outbreaks in wild boars and the incidence in domestic pigs have continued to rise annually [18]. In Bosnia and Herzegovina, specifically in the Republic of Srpska, there are no studies concerning viral diseases in wild boar. The presence of PRV has been confirmed indirectly through infections in hunting dogs [19]. To date, there have been no reports of PCMV or APPV in the region. CSF has not been detected for a decade, largely due to strict vaccination protocols implemented in domestic pig populations [20]. ASF, however, was first reported in 2023, and the number of cases has been steadily increasing since [21]. In both Serbia and the Republic of Srpska, active and passive surveillance programs are in place for the detection of ASF and CSF in dead or diseased wild boars. In Serbia, passive serological surveillance for PRV is also conducted using a subset of samples originally collected for ASF and CSF monitoring. However, the presence of other viral pathogens in wild boar populations, such as PCV2, PPV, APPV, or PCMV, is not currently addressed under official government surveillance or legislative frameworks [22]. Previous studies on PPV, PCV2, and PCMV in Serbia have primarily focused on the eastern parts of the country, with little to no information available from regions bordering the Republic of Srpska [3,14,23]. These bordering areas are of particular interest, given their geographical continuity and the frequent movement of wildlife across national boundaries. Despite the recognized role of wild boars as reservoirs and potential vectors of viral pathogens, recent data on their infection status in the Republic of Srpska are lacking. This limits our understanding of the epizootiological situation in this shared ecosystem and the potential risks it poses to domestic pig populations. The aim of this study is to assess the prevalence and spatial distribution of selected viral pathogens in wild boar populations in Serbia and the Republic of Srpska (Bosnia and Herzegovina). By generating updated data and exploring co-infection patterns, the study seeks to contribute to a better understanding of pathogen dynamics at the wildlife-livestock interface. These insights are crucial for designing effective disease surveillance, prevention, and control strategies, particularly in regions where backyard farming and low biosecurity increase the risk of disease spillover.

## 2. Materials and Methods

Samples for this study were collected from one administrative district (Kolubarski) in the western part of Serbia, as well as from the Republic of Srpska (Bosnia and Herzegovina). Spleen and kidney tissues were collected from healthy wild boars legally hunted during a population reduction program for ASF control. As sampling depended on carcass availability, a non-random convenience sampling approach was employed. Samples were collected from forested areas over a six-month period. After the hunt, the carcasses were transported to refrigerated holding facilities, where samples for Trichinellosis testing were taken and exenteration of the internal organs was performed. All samples were transported to the Institute of Veterinary Medicine of Serbia under cold chain (4 °C) conditions, with no other preservatives, to preserve their integrity. A total of 66 samples were collected, 33 from Serbia and 33 from the Republic of Srpska. All wild boar samples were subsequently subjected to further testing as outlined below.

For each animal, age, sex, weight, location, and date of sample collection were recorded. Age was estimated based on dentition, and animals were categorized according to the following scheme: those without molars were considered to be juveniles under 18 months of age; animals with one molar were classified as subadults 1.5 to 2.5 years old; and those with two molars were considered adults older than 2.5 years, according to the monitoring plan for CSF and ASF in wild boar recommendations [24]. To assess associations between infection status and host factors (sex, age group, and region), Chi-square tests were used for variables with more than two categories, and Fisher’s exact tests were applied when expected cell counts were small. The same tests were used to evaluate associations between co-infection patterns (double infections: PPV + PCV2, PPV + PCMV, PCV2 + PCMV; and triple infection: PPV + PCV2 + PCMV) and host factors (sex, age group, and region). Statistical significance was set at *p* < 0.05. All analyses were performed using Excel (Microsoft Office).

Tissue samples were homogenized using a mortar and pestle in a 1:10 ratio with phosphate-buffered saline (PBS) and then centrifuged at 4000 rpm for 10 min. The supernatant was carefully decanted and stored at −20 °C until further use. Nucleic acid extraction was performed using the IndiMag Pathogen Kit (Indical Bioscience GmbH, Leipzig, Germany), following the manufacturer’s instructions. For the detection of PPV, PCV2, PCMV, PRV, pestiviruses, ASFV, and CSFV, specific sets of primers were used as listed in Table 1. Two different master mixes were employed depending on the type of virus: Luna Universal Probe RT-qPCR Master Mix (New England BioLabs, Ipswich, MA, USA) for RNA viruses and Luna Universal Probe qPCR Master Mix (New England BioLabs, Ipswich, MA, USA) for DNA viruses. For RNA viruses, each 20 μL reaction contained 10 μL of Luna Universal One-Step Reaction Mix (2X), 1 μL of Luna WarmStart^®^ RT Enzyme Mix (20X) (New England BioLabs, Ipswich, MA, USA), 0.8 μL of 10 μM forward primer, 0.8 μL of 10 μM reverse primer, and 5 μL of RNA template, and nuclease-free water to bring the total volume to 20 μL. The thermal cycling protocol included reverse transcription at 55 °C for 10 min, initial denaturation at 95 °C for 1 min, followed by 40–45 cycles of 95 °C for 10 s and 60 °C for 30 s (with fluorescence acquisition), and a final melt curve analysis from 60 °C to 95 °C. For DNA viruses, each 20 μL reaction included 1 μL of 10 μM forward primer, 1 μL of 10 μM reverse primer, 0.5 μL of 10 μM probe, 5 μL of DNA template, and RNA-free water to complete the volume. The thermal cycling conditions consisted of an initial denaturation at 95 °C for 1 min, followed by 50 cycles of 95 °C for 15 s and 60 °C for 30 s. An external VetMax™ Xeno™ DNA/RNA internal positive control (Applied Biosystems, Beverly, MA, USA) was included in each sample to verify the success of nucleic acid extraction.

## 3. Results

Of the 66 wild boar samples analyzed, 57 (86.4%, 95% CI: 76.2–92.8%) were positive for at least one of the three tested pathogens (PPV, PCV2, or PCMV). Specifically, 21 animals (31.8%, 95% CI: 21.7–44.0%) were positive for a single virus, 28 animals (42.4%, 95% CI: 30.6–54.9%) were positive for two viruses, and 8 animals (12.1%, 95% CI: 6.1–22.3%) were positive for all three. Nine animals (13.6%, 95% CI: 7.2–23.8%) tested negative for all viruses included in the panel. Virus-specific prevalence for PPV, PCV2, and PCMV was calculated across the total sample set and stratified by sex, age group, and co-infection rates, as shown in Table 2. A statistically significant association (*p* < 0.05) was found between sex and the occurrence of triple infections (PPV + PCV2 + PCMV) (*p* = 0.0045), whereas no statistically significant association was observed between triple infection status and age group (*p* = 0.3345). The geographic distribution of virus positivity and co-infections by region is presented in Table 3. No statistically significant associations were found between general co-infection status (any two viruses) and sex, age group, or region, although the association between PCV2 + PCMV co-infection and age group approached significance (*p* = 0.0547). Sample distribution is depicted in Figure 1.

## 4. Discussion

Wild boars are recognized as important carriers and reservoirs of infectious diseases, complicating eradication efforts and maintaining viral pockets that pose a continuous risk of spillover into domestic pig populations. In the present study, we analyzed 66 legally culled wild boars for the presence of selected swine viral pathogens. A cumulative prevalence of 86.4% was observed, with 57 out of 66 animals testing positive for at least one pathogen. The highest prevalence was recorded for PCMV at 74.2%, followed by PCV2 at 50%, and PPV at 28.8%. Previous studies on PCMV prevalence in Serbia by Jezdimirović et al. reported a prevalence of 8%, suggesting a marked increase when compared to our findings. However, the samples in that study were collected in 2023, making a rapid nationwide spread of the virus unlikely within such a short timeframe. Additionally, the possibility of sampling bias cannot be excluded, given the relatively small sample sizes involved. Our findings are consistent with those reported in other countries. Studies from Argentina, Italy, and Germany have documented PCMV prevalence rates ranging from 54% to 82%, with specific reports of 56% in Argentina and 54–82% in different regions of Italy and Germany [31,32].

The study by Jezdimirović et al. reported a PPV prevalence of 56%, considerably higher than the 28.8% observed in the present study [14]. Additionally, while Jezdimirović et al. detected PRV in 18% of tested wild boars, no PRV-positive animals were identified in our sample set. Similarly, Nišavić et al. reported a PPV prevalence of 56.6%, further exceeding the levels observed in this study [14,23]. In neighboring Croatia, a seroprevalence of 41.6% for PPV was detected, supporting the notion that higher prevalence rates are common in the broader region [33]. PRV establishes latent infection in the trigeminal ganglia and is typically detectable in blood and visceral organs only during the acute phase, which may account for the low molecular detection rates despite high seroprevalence observed in some studies.

Regarding PCV2, Šolaja et al. reported a prevalence of 56.8%, while Nišavić et al. found a prevalence of 40.2%, both results broadly consistent with the 50% prevalence detected in our study [3,23]. Similar PCV2 prevalence rates have been reported in reproductive tissues of wild boar populations from southern Italy (Campania region) and Germany, with values of 47.3% and 50.7%, respectively [34,35]. In contrast, a study from the Basilicata region of Italy reported a significantly lower prevalence of 27%, highlighting regional variation within the same country [36]. Lower prevalence rates have also been documented elsewhere in Europe, including 20.5% in Hungary, 13.5% in Romania, and 31.8% in Ukraine, further supporting the existence of geographical differences in PCV2 circulation among wild boar populations [37,38,39]. These variations in reported prevalence across studies from Serbia likely reflect underlying geographical differences, as the various studies sampled wild boar populations from distinct ecological regions of Serbia, including the eastern and northern parts of the country, characterized by differing epidemiological pressures, management practices, and landscape features. To date, no published studies have reported on the prevalence of viral diseases in wild boar populations in Bosnia and Herzegovina, including the entity of the Republic of Srpska. However, limited data are available from clinical samples of extensively managed domestic pigs, in which PCV2 and PPV were detected in 15% and 12.5% of cases, respectively [40].

Co-infections were frequent in the examined wild boar population, with 42.4% of animals testing positive for two viruses and 12.1% for all three. The most common co-infection was between PCV2 and PCMV, detected in 42.4% of animals from the Republic of Srpska and 36.4% from Serbia. The second most prevalent was the combination of PPV and PCMV, found in 24.2% of animals from both regions. The least common was the co-infection between PPV and PCV2, observed in 18.2% of animals from the Republic of Srpska and 12.1% from Serbia. In contrast, the study by Jezdimirović et al. reported no co-infections, despite a high prevalence of PPV. A study from Slovakia documented co-infection with porcine parvovirus 3 (PPV3) and PCV2 at 11.8%, while similar rates were observed in China, where PPV and PCV2 co-infections reached 20% [41,42]. Lower rates have been reported elsewhere, such as in India (4.66%), and a study from Japan confirmed PPV and PCV2 co-infection in slaughtered domestic pigs [43,44]. To the best of the authors’ knowledge, this is the first confirmed report of triple co-infection with PPV, PCV2, and PCMV in wild boars. No prior studies have documented co-infection involving PCMV in combination with other viral pathogens in this species. A statistically significant association was observed between triple co-infection status and sex, with 18.2% of males testing positive for all three viruses (PPV, PCV2, and PCMV), compared to only 3.1% of females. Additionally, co-infection with PCV2 and PCMV approached statistical significance, with 60% of subadult animals testing positive for both viruses. These findings may be attributed to ecological and behavioral patterns that influence contact rates within wild boar populations. Subadult wild boars—particularly yearlings—often exhibit the highest contact heterogeneity, as they tend to move between social groups during dispersal, increasing their likelihood of exposure to multiple pathogens [45]. This consideration is particularly important in the context of persistent infections, such as those caused by PCV2 and PCMV. PCV2 is often regarded as a commensal pathogen under subclinical conditions, while PCMV is known to establish lifelong latent infections, with potential reactivation under stress or immunosuppression [46,47]. Similar findings were reported by Šolaja et al., who observed a higher prevalence of infection in male wild boars [3]. However, other studies have not demonstrated a consistent association between sex and infection status, suggesting that sex-related differences may vary depending on population structure, environmental factors, or sampling design [14,33,48,49]. The absence of ASF and CSF in this study is expected. CSF has not been reported in either country for over a decade and is typically self-limiting in wild boar [20]. ASF has not been detected in wild boar in the sampled region of Serbia, and its high fatality rate often results in low observed prevalence despite environmental persistence [50].

The high detection rates observed in this study may also be influenced by the choice of tissue sampled. Spleen tissue, which was used for viral detection, is a known target for both PCV2 and PCMV replication [51]. In addition to host and population-level factors, the type of tissue analyzed may have affected the prevalence estimates, as the spleen is a primary site of viral replication and persistence for both PCV2 and PCMV, which may contribute to higher detection rates compared to other tissues.

In both countries, pig production is largely dominated by backyard farms, where biosecurity is minimal or absent [2]. This creates favorable conditions for the bidirectional transmission of pathogens between wild boars and domestic pigs. In rural areas, it remains common for domestic pigs to graze outdoors during warmer months, increasing the likelihood of direct or indirect contact with wild boars. Moreover, in certain regions, the intentional breeding of wild boar–dominant pig hybrids for culinary purposes, considered a delicacy, further exacerbates the risk of cross-species transmission [3,4,52]. Moreover, many backyard farmers are employed in large commercial farms, creating a potential route for disease introduction into high-biosecurity systems. A similar pattern was observed during the ASF outbreaks in 2023 [18]. Although PCMV has not yet been reported in Serbia’s domestic pig population, the high prevalence in wild boars and the link to low-biosecurity farms suggest an elevated risk of spillover.

Current surveillance of wild boar in both Serbia and the Republic of Srpska focuses primarily on ASF and CSF, with additional PRV serological testing conducted in Serbia to assess transmission risk to domestic pigs. In domestic herds, PCV2 and PPV are typically included in diagnostic panels for reproductive disorders. There are no government-mandated eradication programs for the investigated pathogens, except for ASF and CSF, which are subject to official control measures in both domestic pigs and wild boar [24]. For PRV, eradication programs exist but are implemented voluntarily in commercial pig farms [53]. In contrast, PCMV is not routinely monitored in either wild boar or domestic pigs. Given its high prevalence, frequent co-infection with PCV2 and PPV, and potential for spillover, particularly via low-biosecurity backyard farms that may act as a bridge to commercial operations, PCMV should be incorporated into national molecular surveillance programs. This is especially relevant considering its recognized zoonotic potential through xenotransplantation [46].

This study is limited by its relatively small sample size and restricted geographic coverage, which may not fully reflect the variability of viral circulation across Serbia and the Republic of Srpska. The sample size reflects the availability of carcasses collected during official ASF control efforts, which constrained both the number and origin of animals. Despite this, significant associations and triple co-infections were detected, suggesting meaningful trends. The absence of molecular characterization also limits insight into strain diversity, recombination, and transboundary transmission. Future studies should include broader sampling and molecular analyses to understand viral evolution and spread better.

## 5. Conclusions

This study documents a high prevalence of PCMV, PCV2, and PPV in wild boar populations from western Serbia and the Republic of Srpska, with frequent co-infections and the first reported case of triple infection involving all three viruses. The findings point to regional variation in viral circulation, potentially influenced by ecological and behavioral factors, as well as differing farm management practices. Observed differences in infection by sex and age further illustrate the complex dynamics of pathogen spread in wild boar populations. Given the high detection rates and potential for spillover, particularly through low-biosecurity backyard farms, greater attention to PCMV within national surveillance frameworks may be warranted. Integrating molecular characterization into future studies could also enhance understanding of strain diversity and transmission pathways.

## Figures and Tables

**Figure 1 pathogens-14-00710-f001:**
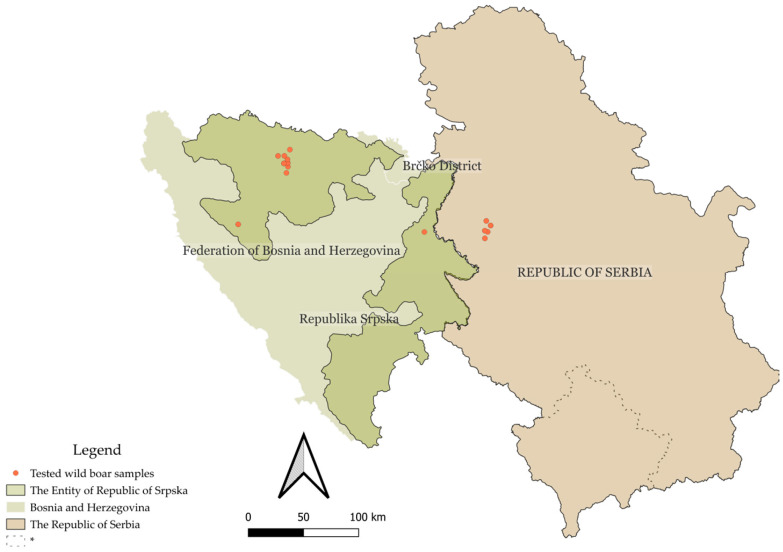
Geographic map of the Republic of Serbia and Bosnia and Herzegovina, with the territory of the Republic of Srpska highlighted in green. Sample collection sites are indicated by red dots. * This designation is without prejudice to positions on status and is in accordance with United Nations Security Council Resolution 1244 (1999) and the advisory opinion of the International Court of Justice on the Kosovo Declaration of Independence.

**Table 1 pathogens-14-00710-t001:** List of primers used in the study.

Primers	Reference
PPVF: CCAAAAATGCAAACCCCAATA PPVR: TCTGGCGGTGTTGGAGTTAAG Probe: Tamra—CTTGGAGCCGTGGAGCGAGCC—Fam	[25]
PCV2F: ATTACCAGCGCACTTCGG PCV2R: GGGTCCGCTTCTTCCATT Probe: Tamra—AGCAGCAACATGCCCAGCAAGAAG—Fam	[26]
PRV gB718F: ACAAGTTCAAGGCCCACATCTAC PRV gB812R: GTCYGTGAAGCGGTTCGTGAT PRV gB785P: Tamra—ACGTCATCGTCACGACC—Fam	[27]
PCMVF: GCTGCCGTGTCTCCCTCTAG PCMVR: ATTGTTGATAAAGTCACTCGTCTGC Probe: Tamra—CCATCACCAGCATAGGGCGGGAC—Fam	[28]
ASFF: CTGCTCATGGTATCAATCTTATCGA ASFR: GATACCACAAGATCRGCCGT Probe: Tamra—CCACGGGAGGAATACCAACCCAGTG—Fam	[29]
Panpesti BVD 190-F: GRAGTCGTCARTGGTTCGAC Panpesti ML 121: TCAACTCCATGTGCCATGTAC Probe: Tamra—TGCYAYGTGGACGAGGGCATG—Vic	[30]

**Table 2 pathogens-14-00710-t002:** Prevalence of PPV, PCV2, and PCMV, as well as their respective co-infections (PPV + PCV2, PPV + PCMV, PCV2 + PCMV, PPV + PCV2 + PCMV), in wild boar populations stratified by age group (Juvenile, Subadult, and Adult) and sex (Male, Female). Results are presented as the number and percentage of positive animals out of the total number of tested samples (n = 66).

Category	Total (n = 66)	Total %	Juvenile (%)	Subadult (%)	Adult (%)	Male (%)	Female (%)
PPV	19	28.8	35.7	13.3	30.4	30.3	27.3
PCV2	33	50	53.6	66.7	34.8	42.4	57.6
PCMV	49	74.2	75	93.3	60.9	78.8	69.7
PPV + PCV2	10	15.2	21.4	13.3	8.7	18.2	12.1
PPV + PCMV	16	24.2	32.1	13.3	21.7	30.3	18.2
PCV2 + PCMV	26	39.4	42.9	60	21.7	36.4	42.4
PPV + PCV2 + PCMV	8	12.1	7.5	3.03	1.5	18.2 *	3.1

* Statistically significant difference (*p* = 0.0045).

**Table 3 pathogens-14-00710-t003:** Prevalence of PPV, PCV2, PCMV (previously known as PCMV), and their co-infections (PPV + PCV2, PPV + PCMV, PCV2 + PCMV, and PPV + PCV2 + PCMV) in wild boar samples (n = 66), stratified by region (Republic of Srpska and Serbia). Results are presented as the number and percentage of positive samples relative to the total number tested and separately for each region.

Category	Total (n = 66)	Total (%)	Republic of Srpska	Serbia
PPV	19	28.8	9 (27.3%)	10 (30.3%)
PCV2	33	50	18 (54.5%)	15 (45.5%)
PCMV	49	74.2	25 (75.8%)	24 (72.7%)
PPV + PCV2	10	15.2	6 (18.2%)	4 (12.1%)
PPV + PCMV	16	24.2	8 (24.2%)	8 (24.2%)
PCV2 + PCMV	26	39.4	14 (42.4)	12 (36.4%)
PPV + PCV2 + PCMV	8	12.1	5 (62.5%)	3 (37.5%)

## Data Availability

All data is available upon request from the corresponding author.

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
