# Peer review of "First Report of Triple Viral Co-Infection (PPV, PCV2, PCMV) in Wild Boars in the Western Balkans"

_pathogens, 2025, doi:10.3390/pathogens14070710_

Round 1
Reviewer 1 Report
Comments and Suggestions for Authors
The manuscript is well-written and describes results which are of general agricultural interest. The writing style is succinct and the presentation quality of the figures and graphs is high.
The introduction covers the background material of the subject in a comprehensive manner
and the work of others in the field, as well as the associated technical design for the current
study, are covered thoroughly. The methods are described in sufficient detail for any aspect of the study to be repeated by others if necessary and the statistical analyses which have been applied to the resulting data seem to be rigorous. The research findings are interesting, particularly the association between triple infection and gender in young animals and would seem to have a strong leaning on farming pigs globally. Rightly, the basis for these findings is discussed in some depth in the conclusions.
This referee has a structural biology background and is therefore not a specialist in this subject.
However, subject to satisfactory additional refereeing, I would have no hesitation in recommending publication of the manuscript following attention to the very minor point(s) below.
On lines 54 to 56 some of the detailed molecular biology of one of the viruses (PRV) is described. For consistency, maybe a similar sentence should be given for the other viruses mentioned in the study or maybe this sentence could be omitted.
Author Response
I would like to express my sincere gratitude for your valuable feedback and thoughtful suggestions during the review process. Your insights have greatly contributed to improving the quality of my paper. I appreciate the time and effort you have invested in providing constructive comments, and I am truly grateful for your guidance.
Thank you for your support and for helping to refine this work.
Comments given by Reviewer 1:
- On lines 54 to 56 some of the detailed molecular biology of one of the viruses (PRV) is described. For consistency, maybe a similar sentence should be given for the other viruses mentioned in the study or maybe this sentence could be omitted.
RESPONSE
- As the reviewer suggested, the following lines were omitted.
Reviewer 2 Report
Comments and Suggestions for Authors
the manuscript reports the results of the testing for several porcine-related viruses on legally hunted boars. The study relies on molecular techniques to screen for six viruses on visceral tissues of legally hunted boars in the western Balkan territories. The study provides important epidemiological data of high interest for veterinary and public health, policymakers, and for hunting management. That said, there are a few aspects of the manuscript that require attention to adequate the manuscript for publication. I provide my comments and suggestions here below:
Introduction
lines 70-71: please, adequate the reference of Jezdimirović et al. (2023). To adequate it to the format of the journal, remove the "(2023)", as it is cited normally at the end of the sentence.
Methods
Methods could benefit from a reorganization into three main sections
- Area of study and characterization of sampling: better characterize the areas where the samples were obtained (whether areas are forested, agricultural, urban...). Characterize the sampling regime: what was the timespan for sampling for each area, how were the samples obtained (butchering, dissection in lab...), more information on the cold-chain storage used (refrigeration, freezing, if other preservatives were used).
- Characterization of the sampled individuals: how did the information of each boar was obtained, with adequate references for the methods (ageing based in dentition)
- Preparation of samples and testing: how were samples pre-processed, how were the molecular tests were performed (this section is adequate as presented in the manuscript)
- Statistical analyses: how the tests for associations between coinfection and sample characteristics.
lines 118-120: the authors mention cell counts, with no previous mention in the methods. Please explain what cells are being referred to.
Table 1: remove the reference citations presented as Name et al. (date), as they are already presented in the journal's format.
Results
Table 3: present also the absolute value for positivity per area.
Discussion
lines 187-188, 196, 199, 206, 230, 251: please, adequate the reference to the journal's style.
Author Response
I would like to express my sincere gratitude for your valuable feedback and thoughtful suggestions during the review process. Your insights have greatly contributed to improving the quality of my paper. I appreciate the time and effort you have invested in providing constructive comments, and I am truly grateful for your guidance.
Thank you for your support and for helping to refine this work.
Comments given by Reviewer 1:
- lines 70-71: please, adequate the reference of Jezdimirović et al. (2023). To adequate it to the format of the journal, remove the "(2023)", as it is cited normally at the end of the sentence.
- Area of study and characterization of sampling: better characterize the areas where the samples were obtained (whether areas are forested, agricultural, urban...). Characterize the sampling regime: what was the timespan for sampling for each area, how were the samples obtained (butchering, dissection in lab...), more information on the cold-chain storage used (refrigeration, freezing, if other preservatives were used).
- Characterization of the sampled individuals: how did the information of each boar was obtained, with adequate references for the methods (ageing based in dentition)
- Preparation of samples and testing: how were samples pre-processed, how were the molecular tests were performed (this section is adequate as presented in the manuscript)
- Statistical analyses: how the tests for associations between coinfection and sample characteristics.
- Statistical analyses: how the tests for associations between coinfection and sample characteristics.
- Table 1: remove the reference citations presented as Name et al. (date), as they are already presented in the journal's format.
- Table 3: present also the absolute value for positivity per area.
- lines 187-188, 196, 199, 206, 230, 251: please, adequate the reference to the journal's style.
RESPONSE
- Corrected as the reviewer suggested.
- The following sentences were added in the manuscript: “Samples were collected from forested areas over a six-month period. After the hunt, the carcasses were transported to refrigerated holding facilities, where samples for Trichinellosis testing were taken and exenteration of the internal organs was performed. All samples were transported to the Institute of Veterinary Medicine of Serbia under cold chain (4 °C) conditions, with no other preservatives, to preserve their integrity.“
- Ageing was conducted based on the following reference, which has been included in the manuscript as per the reviewers' recommendations. “Plan and Method of Monitoring for Classical Swine Fever and African Swine Fever in Wild Boars in 2024 and 2025” given out by the Ministry of Agriculture, Forestry and Water Management, Veterinary Directorate as a part of the Monitoring for CSF and ASF.
- Nothing adapted
- And 6.
The following sentence was added according to the reviewers' comments explaining how the tests were conducted.
“To assess associations between infection status and host factors (sex, age group, and region), Chi-square tests were used for variables with more than two categories, and Fisher’s exact tests were applied when expected cell counts were small. The same tests were used to evaluate associations between co-infection patterns (double infections: PPV+PCV2, PPV+PCMV, PCV2+PCMV; and triple infection: PPV+PCV2+PCMV) and host factors (sex, age group, and region). Statistical significance was set at p < 0.05. All analyses were performed using Excel (Microsoft Office).”
- Corrected as per the reviewers' recommendations.
- Corrected as per the reviewers' recommendations.
- Corrected as per the reviewers' recommendations.
Reviewer 3 Report
Comments and Suggestions for Authors
This article describes the molecular detection of some swine pathogens in wild boar organs (spleen) from two European countries. The work is well written, matches the aims of the journal, the methodology used is rigorous (although no sequencing or particular analyses were performed). The big weak point concerns the sampling, which in two countries where wild boar is widespread, cannot be only 66 animals. My suggestion is to convert the work into a short and to address the following comments:
1) Line 18: Is there an eradication plan for these pathogens in pigs? Discuss in the discussion section.
2) Line 23: Try to explain these data. Furthermore, there is no evidence in the manuscript of these results (table?)
3) Line 110: How was it calculated? Was it a convenience sampling?
4) Line 117: Add reference. Results: Chi-square table missing
5) Line 209: The studied pathogens were also found in reproductive tissues of wild boars: Detection of selected pathogens in reproductive tissues of wild boars in the Campania region, southern Italy
6)Discussion: The authors should specify that the choice of the sample, in addition to other variables, can influence the prevalence found. The spleen, for example, should ensure obtaining high prevalences.
Author Response
I would like to express my sincere gratitude for your valuable feedback and thoughtful suggestions during the review process. Your insights have greatly contributed to improving the quality of my paper. I appreciate the time and effort you have invested in providing constructive comments, and I am truly grateful for your guidance.
Thank you for your support and for helping to refine this work.
Comments given by Reviewer 3:
- This article describes the molecular detection of some swine pathogens in wild boar organs (spleen) from two European countries. The work is well written, matches the aims of the journal, the methodology used is rigorous (although no sequencing or particular analyses were performed). The big weak point concerns the sampling, which in two countries where wild boar is widespread, cannot be only 66 animals. My suggestion is to convert the work into a short and to address the following comments:
- Line 18: Is there an eradication plan for these pathogens in pigs? Discuss in the discussion section.
- Line 23: Try to explain these data. Furthermore, there is no evidence in the manuscript of these results (table?).
- Line 110: How was it calculated? Was it a convenience sampling?
- Line 117: Add reference. Results: Chi-square table missing
- Line 209: The studied pathogens were also found in reproductive tissues of wild boars: Detection of selected pathogens in reproductive tissues of wild boars in the Campania region, southern Italy
- Discussion: The authors should specify that the choice of the sample, in addition to other variables, can influence the prevalence found. The spleen, for example, should ensure obtaining high prevalences.
RESPONSE
- Thank you very much for your valuable comments. With regard to the suggestion of converting the manuscript into a Short Communication, we would like to note that the first reviewer recommended publication with only minor revisions, and the second reviewer did not address this issue. Therefore, we will proceed with revising the manuscript as a regular article, unless the editor specifically requests its conversion to a Short Communication, as both reviewers provided only minor comments on the current format.
- The following sentence was added in the discussion section as per the reviwers comments “There are no government-mandated eradication programs for the investigated pathogens, except for ASF and CSF, which are subject to official control measures in both domestic pigs and wild boar [24]. For PRV, eradication programs exist but are implemented voluntarily in commercial pig farms [53].”
- The data in line 23 “Subadult wild boars showed the highest PCV2 + PCMV co-infection rate (p = 0.0547).” is given in the results section under the lines 163-166 “No statistically significant associations were found between general co-infection status (any two viruses) and sex, age group, or region, although the association between PCV2 + PCMV co-infection and age group approached significance (p = 0.0547).”
- The number of samples analyzed reflects the total number of specimens that were available to us through collaboration with the hunting associations during the study period.
- Reference added. The chi-squared results have been added as per the reviewer's suggestion.
- Line 209. Corrected as proposed by the reviewer.
- A sentence added as proposed by the reviewer, lines 262-265 “In addition to host and population-level factors, the type of tissue analysed may have affected the prevalence estimates, as the spleen is a primary site of viral replication and persistence for both PCV2 and PCMV, which may contribute to higher detection rates compared to other tissues”
Round 2
Reviewer 3 Report
Comments and Suggestions for Authors
I thank the authors for addressing some of my comments, however the most important ones which represent the weaknesses of the article, were not addressed.
Author Response
Dear Reviewer,
We apologise for not addressing the comment regarding the sample size in the previous revision.
I would like to sincerely thank you for your valuable feedback and thoughtful suggestions throughout the review process. Your insights have significantly contributed to improving the quality of this manuscript. I appreciate the time and effort you dedicated to providing constructive comments and guidance. Thank you again for your support and for helping us strengthen this work.
Comments given by Reviewer 3:
- I thank the authors for addressing some of my comments, however the most important ones which represent the weaknesses of the article, were not addressed.
RESPONSE
We fully acknowledge that the number of wild boars included in this study (n = 66) represents a limitation, particularly given the broad distribution and high population density of wild boar in both Serbia and the Republic of Srpska. However, this study was designed as a preliminary investigation to assess the molecular prevalence and co-infection patterns of understudied viral pathogens, specifically PCMV, PCV2, and PPV, in a region lacking recent or any data on these infections, particularly in the Republic of Srpska. Samples were collected under the framework of legal population control programs primarily aimed at ASF monitoring, which constrained both the number and geographic spread of the specimens. Nonetheless, the sample size was sufficient to reveal statistically significant differences in co-infection patterns by sex and age, and to detect for the first time in the Western Balkans a triple co-infection involving PCMV, PCV2, and PPV in wild boars. These findings provide an important baseline and highlight the need for larger-scale, systematic surveillance efforts. We have added a clarification of this limitation and its implications for interpretation to the Materials and Methods section (Lines 104-106) and the Discussion section (Lines 296–304).
Lines 104-106: “Spleen and kidney tissues were collected from healthy wild boars legally hunted during a population reduction programme for ASF control. As sampling depended on carcass availability, a non-random convenience sampling approach was employed.”
Lines 209-304: “This study is limited by its relatively small sample size and restricted geographic coverage, which may not fully reflect the variability of viral circulation across Serbia and the Republic of Srpska. The sample size reflects the availability of carcasses collected during official ASF control efforts, which constrained both the number and origin of animals. Despite this, significant associations and triple co-infections were detected, suggesting meaningful trends. The absence of molecular characterisation also limits insight into strain diversity, recombination, and transboundary transmission. Future studies should include broader sampling and molecular analyses to understand viral evolution and spread better.”
Round 3
Reviewer 3 Report
Comments and Suggestions for Authors
The authors have addressed the previous comments
Author Response
Dear Reviewer,
Thank you for your comments.
Kind regards.